# The Development of the State Emblems and Coats of Arms in Southeast Europe

**Jovan Jonovski** [ID]

Macedonian Heraldic Society, 1000 Skopje, North Macedonia; jonovski@gmail.com

**Abstract:** Heraldic traditions in southeast European countries are similar, as are the histories of their state emblems and coat of arms. Their development could be classified into three periods: (1) from the founding of the states until the end of World War II; (2) the socialist period; and (3) the period of democratisation after the collapse of socialism. The focus of this work is the processes of the adoption of coats of arms. The descriptions are taken from the appropriate legal documents. This paper examines the emblems and coats of arms of modern southeastern European, or Balkan states, Albania, Bosnia and Herzegovina, Bulgaria, Croatia, Greece, Kosovo, Macedonia, Montenegro, Romania, Slovenia, and Serbia.

**Keywords:** Balkan heraldry; national heraldry; states coat of arms

## 1. Introduction

The term southeast Europe is a euphemism for the Balkans which, although a geographical one, has more of a culturological meaning. According to the Encyclopaedia Britannica, Balkanization "explains the disintegration of some multiethnic states and their devolution into dictatorship, ethnic cleansing, and civil war." (Pringle 2010). Although there are no unique clear borders of the Balkans, we will use the borders of the rivers Soča–Sava–Danube. The deltas of these rivers also border a very small part of the territory of Italy and Ukraine, but they are not considered Balkan countries. In this paper, the development of the emblems and coats of arms of the following countries, Albania, Bosnia and Herzegovina, Bulgaria, Croatia, Greece, Kosovo, Macedonia, Montenegro, Romania, Slovenia, and Serbia will be discussed.

The coat of arms[1], the flag, and the anthem are national symbols that have a certain role inside and outside of the country. The primary purpose of state symbols, on the internal level, is to symbolise the state and to form a common identity by creating an agreement on the basic meaning of the symbols among citizens. This identity is maintained by the elites, with continuous monitoring, and in special circumstances, involves the citizens in renegotiating or confirming the existing meanings (Poljak 2013). Externally, state symbols represent the state (or connection with the state) to other states and their citizens.

With the state coat of arms, the armiger is the state, and it represents the state; that is, the state apparatus, the administration on a defined territory. Historically, coats of arms in Europe, for the most part, were either assigned by a ruler or simply taken from the coat of arms of the ruler of a territory that was part of the state. Often, the ruling dynasties changed, but the coat of arms remained the same. Or, the new ruler took the coat of arms of the territory as his own or supplemented his coat of arms with the coat of arms of the territory or vice versa. It is called the Arms of Dominion. From a sign of visual identity of individuals, by marking properties and possessions, the coat of arms turns into territorial heraldry from where it can be transferred to the entire country (Јоновски 2015, p. 111).

The choice of the emblem depends on a social context that favours some symbols over others.[2] Then, when those socio-political relations change, the attitude towards specific symbols and their conditional hierarchical arrangement changes. Therefore, often during

the change in these relations, symbols that were suppressed are exalted as stronger symbols, and that suppression is sometimes perceived as giving a greater weight to those symbols, in the narration as an additional symbol for the new social reality.

The symbolic design of the national/state identity is a product of specific, socio-political events that surround the adoption of the symbol, which is first accepted by a group of people—an elite, in a given socio-political context—and it is later transmitted to others.

Harold Laswell, in addition to the category of national symbols chosen to indicate identification with tradition or historical heritage, points out that there are two other categories: preference—freedom, independence, justice; and aspiration or expectations—progress, the inevitability of a world revolution, the proletariat (a symbol of economic class) (Laswell et al. 1952, p. 15).

There are three historical periods that have a major impact on heraldic processes and the state coat of arms of the southeastern European states. The first is the period from the creation of the state to the end of the Second World War. The second begins with the Socialist Revolution of 1944–1945, while the third is the period of democratisation after the fall of the Berlin Wall, from 1989 until today.

## 2. First Period (1821–1944)

This is the period of creation of the states and their state symbols, emblems, or coats of arms. All created coats of arms symbolically indicate historical continuity to respective medieval states, provinces/territories, or great leaders through their actual or attributed coats of arms. The land coats of arms from the Illyrian armorials and the Stemmatographies (Vitezovic 1701 in Latin and Zhefarovic 1741 in Church Slavonic) are attributed coats of arms (coats of arms of territory as an idea, not of a real administration), which, in fact, become real coats of arms only with the formation of the states. All coats of arms mentioned here, except for the coats of arms of Banat and Dobruja, which were created later, are found in the Stemmatography (Jonovski 2020). Only the emblems of Greece deviate here, which have been changed several times, with the first two being symbolically and mythologically connected to ancient Hellada. On the other hand, the coat of arms of the ruling dynasty is also present in most of them, usually in an escutcheon of pretence.

### 2.1. Greece

Greece is the first country in southeastern Europe that was formed and adopted a state emblem with the Epidaurus constitution of 1 January 1822, and the decree of 15 March of the same year; the first coat of arms was adopted, which shows the goddess Athena and an owl (Figure 1a). In 1828, the new Republic of Greece, headed by Ioannis Antonios Kapodistrias, adopted a new state emblem with a phoenix rising from the ashes (a traditional theme in Greek mythology), above which there is a cross (for the Christian faith of the state) and the year 1821 written below (Figure 1b). A blue flag with a white cross was adopted, which later became the basis for the design of the coat of arms (The Flag n.d.).

In 1831, Prince Otto (Wittelsbach) of Bavaria sat on the throne and the coat of arms of Greece became Azure, a cross couped Argent bearing an escutcheon of pretence with the coat of arms of Bavaria, i.e., of the House of Wittelsbach. The shield is crowned with a royal crown and has supporters—crowned lions (Figure 2a). This coat of arms was discarded after the King's expulsion in 1862. When the young Prince William of Denmark was elected king in 1864, the escutcheon of pretence was changed to the dynastic arms of Schleswig–Holstein–Sondenburg–Glücksburg, as were the supporters, two figures of Herakles (Figure 2b). The Order of the Saviour and the motto of the dynasty, Ἰσχύς μου ἡ ἀγάπη τοῦ λαοῦ ("My strength is love for the people") were also added (Јоновски 2015, p. 344).

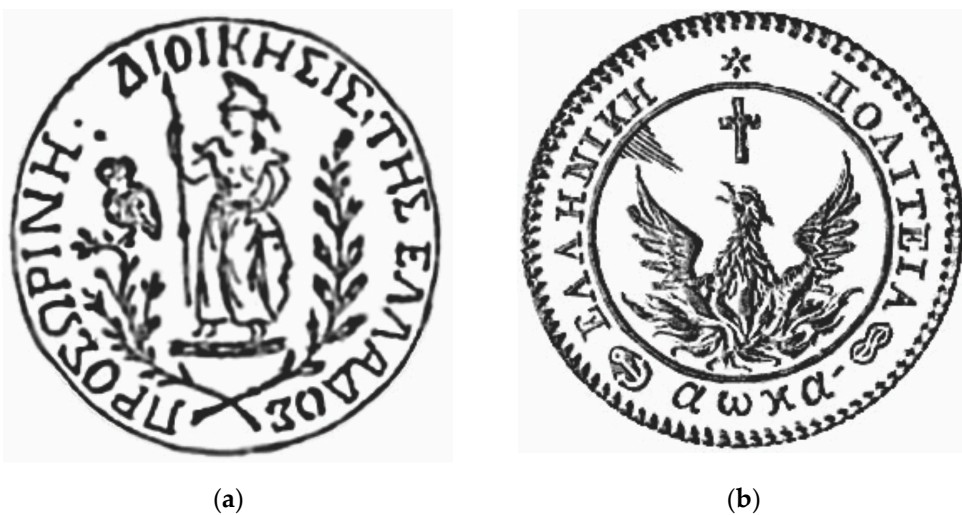

**Figure 1.** State emblems of Greece: (**a**) 1822; (**b**) 1828.

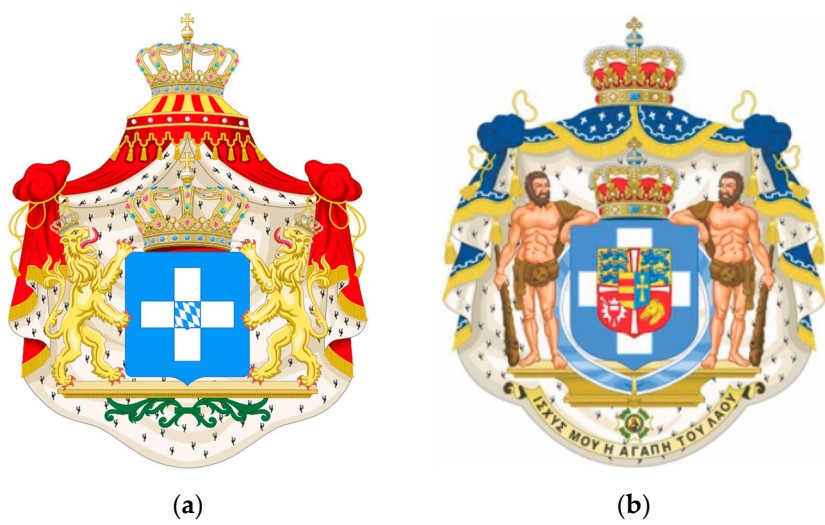

**Figure 2.** Coat of arms of Greece: (**a**) 1831–1862; (**b**) 1864–1935.

When Greece became a republic in 1924, all external elements were discarded, and the escutcheon of pretence was replaced by a double-headed eagle. This coat of arms was in use for only two years. In 1935, the monarchy was restored under the leadership of George II, who had ruled before the proclamation of the republic (Figure 3) (The National Emblem n.d.).

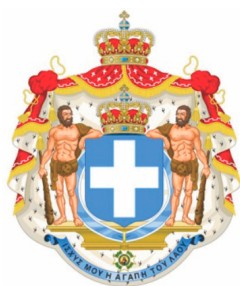

**Figure 3.** Coat of arms of Greece 1935–1967.

*2.2. Serbia*

With the establishment of the Principality of Serbia in 1815, the coat of arms of Serbia from the Stemmatographia of Zhefarovic was taken for the coat of arms, additionally

encircled with olive and oak branches. In the first Serbian Constitution from 1835, the official blazon was given, which said that the coat of arms of Serbia was a white cross on a red field, with silver fire steels between the arms of the cross turned towards the cross. The whole coat of arms is surrounded by a green wreath, on the right by oak trees, and on the left by olive leaves (Figure 4a); Gules, a cross between four furisons Argent (Уставъ 1835).

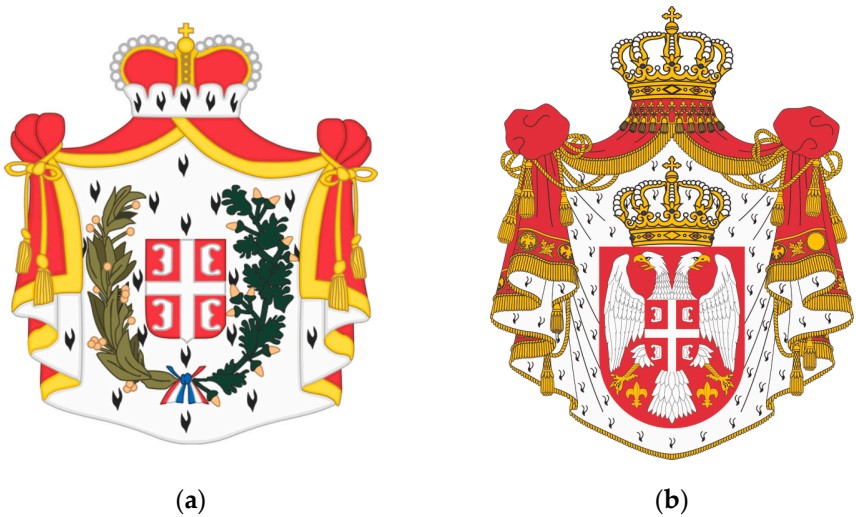

(**a**)　　　　　　　　　　　　　(**b**)

**Figure 4.** Coat of arms of Serbia: (**a**) 1835–1882; (**b**) 1882–1918.

The design with the four furisons (fire steels) comes from the flag of the restored Byzantine Empire where the fire steels actually represent the letter "B", which is probably an acrostic of the motto of the Paelologists, "Βασιλεὺς Βασιλέων Βασιλεύων Βασιλευόντων" translated as King of kings reigns the kings. Over time, these fire steels were perceived as the letter "S" and the four letters are believed to denote the slogan: "Само Слога Србина Спашава" (Only Unity Saves the Serbs).

In 1882, the Kingdom of Serbia adopted a new coat of arms (Закон о грбу Краљевине Србије 1882), from an idea by Stojan Novaković, with the addition of the coat of arms of the medieval ruling dynasty of Nemanjic from the Illyrian Armorial, as a symbol of old, medieval Serbia, to establish a symbolic link of continuity (Круна води републику 2006). The coat of arms was Gules a double-headed eagle argent, surmounted by a royal crown; under each claw a golden lily. On its breast is the coat of arms of the Principality of Serbia: Gules, a cross between four furisons Argent. Above the shield, a royal crown. The design was by Ernst Khrall (Figure 4b).

With the creation of the Kingdom of Serbs, Croats, and Slovenes in December 1918, on the Serbian coat of arms, the escutcheon of Serbia was replaced with per fess dexter per pale (1) Serbia—Gules, a cross between four furisons Argent, (2) Croatia—chequy of Argent and Gules; (3) Slovenia—Azure a star and in base a crescent Argent. The crown was replaced by the so-called Yugoslav crown. In 1921, the Slovenian arms had three stars Or replaced by one Argent (Figure 5a).

During the Second World War, the coat of arms of Serbia, as a puppet state under the rule of Germany, was similar to that of the Kingdom of Serbia, but without a crown on the shield (Figure 5b).

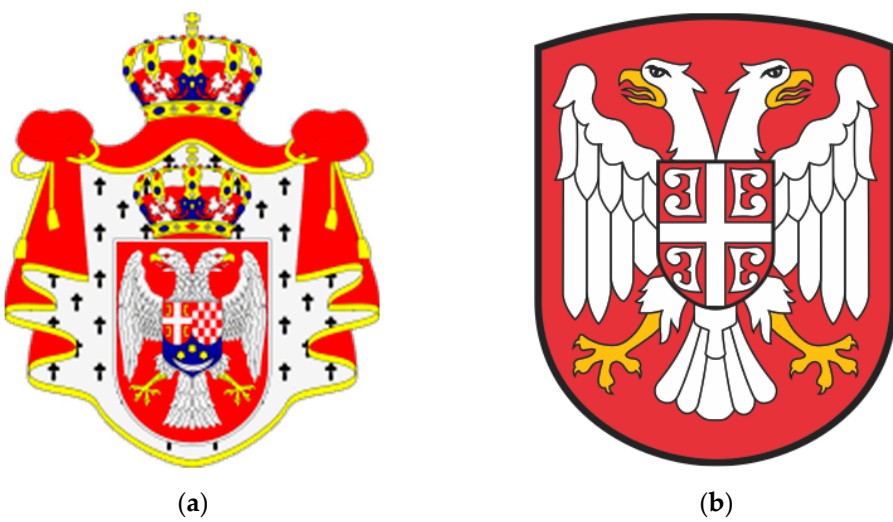

(**a**)

(**b**)

**Figure 5.** Coat of arms: (**a**) Kingdom of Serbs Croats and Slovene, Kingdom of Yugoslavia 1918 (1921)–1941 (**b**) Serbia 1941–1944.

*2.3. Romania*

Romania is mainly situated above the Danube and is a Central European country, both by geography and according to heraldic practice. Romania was originally formed by the unification of the two countries of Wallachia and Moldavia in 1859 as a personal union under Prince Alexander Ioan Cuza. Then, a marshalled coat of arms was used unofficially with different heraldic combinations of the coats of arms of the two countries, but also versions in which the fields of the coats of arms contain the national colours, blue, yellow, and red (Figure 6a). In 1867, Carol I of Hohenzollern–Sigmaringen was elected Prince of the United Principalities of Romania and the following year he issued a decree establishing the coat of arms.

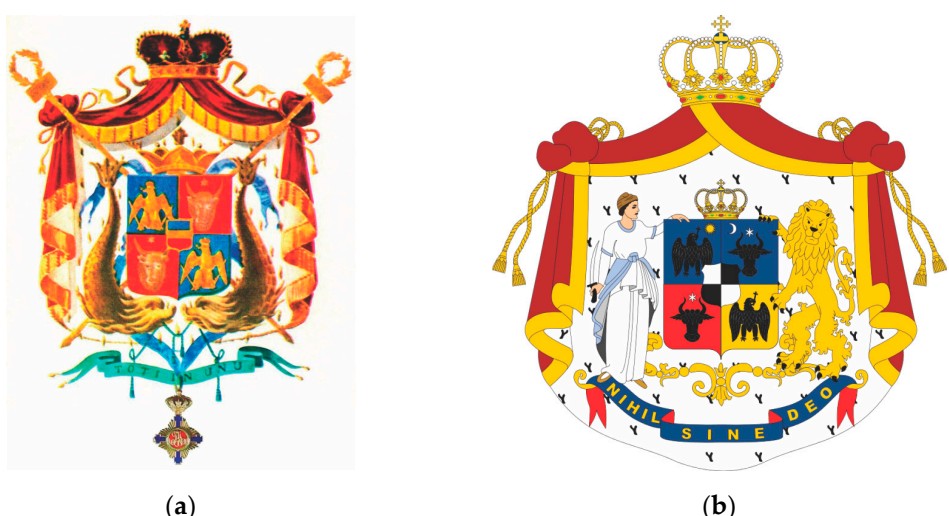

(**a**)

(**b**)

**Figure 6.** Coat of arms of (**a**) Wallachia and Moldova 1859; (**b**) Romania 1867–1872.

Articles 1–4 describe the coat of arms of Romania as quarterly (Figure 6b): (1) (Top right)[3] Azure and (4) (Bottom left) Or, the Romanian eagle with its head to the left wing and a cross Or in its beak, the symbol of Wallachia. On (2) (Top left) Azure and on (3) (Bottom right) Gules, a bull's head with a star between horns with the sun and moon in front, the symbol of Moldavia. There is a royal crown on the shield. On a shield the arms of His Majesty: quartered (1) and (4) Argent, (2) and (3) Sable. Supporters: on the left a lion, and on the right a woman in Dacian costume, holding in her left hand a Dacian weapon,

called a harpy. On the base is a blue ribbon inscribed with the Hohenzollern family motto, "Nihil sine Deo" (Nothing without God). The robe of estate is Gules lined with ermine, with the royal crown on top (Înaltul Decret Domnesc 1867).

With a decree of Carol I from 1872, the coat of arms was supplemented by a change of the third field, to Gules a lion Or issuing from a crown with a star between its forelegs (symbol of Craiova), and the fourth Azure two dolphins urinant respectant Or (symbol of the coastal lands). The right supporter, the lion is also changed. The first field was also redefined, so the eagle was crowned and held a sword and a sceptre and on the right side there is a golden sun (Figure 7a) (Înaltul Decret Domnesc 1872).

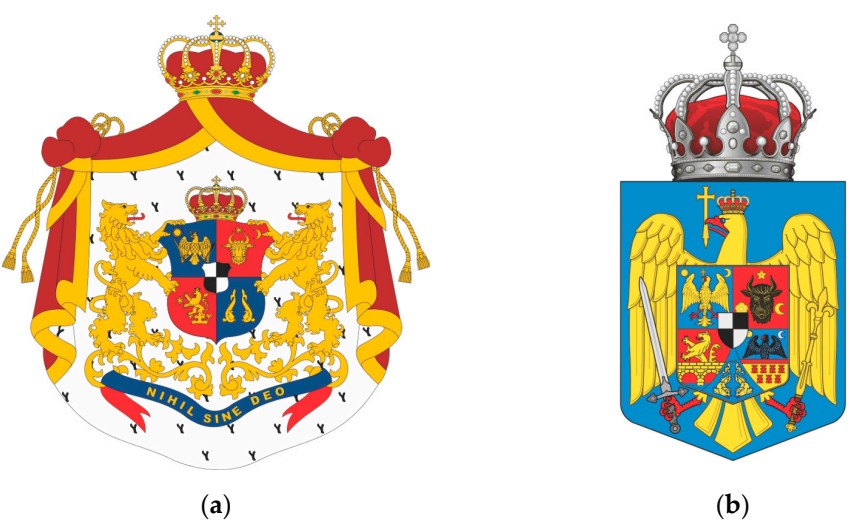

(**a**)  (**b**)

**Figure 7.** Coat of arms of Romania: (**a**) 1872–1921; (**b**) 1921–1947.

In this way, the coat of arms of Romania follows the traditional practice of arms of dominion, where the fields with the coats of arms of the land's ruler are supplemented with the change of territory. In 1921 the number of fields on the Dominion coat of arms was increased, but it was placed on the breast of a golden eagle holding a cross in its beak (Figure 7b). It was defined as a coat of arms consisting of three shields on top of each other. The great shield is Azure, an eagle crowned Or armed Gules, holding in its beak cross pattée fitchée Or, in the right claw a sword, in the left a sceptre with a lily. On the eagle's breast is an escutcheon quarterly and enty in point, "composed of the united sister countries": (1) the coat of arms of Wallachia surmounted by a decrescent Or, (2) Moldavia (with Bessarabia and Bukovina) surmounted by rose (with five leaves) in dexter, crescent Or in sinister. (3) coat of arms of Romanian Banat of Severinus: Gules, over natural waves, a bridge with two arched openings Or, built of carved stone (Trajan's Bridge), from which issues a lion Or. (4) Transylvania (with parts of Krishana and Maramuresh): Per fess Azure and Or, a bar Gules issuant therefrom an eagle displayed Sable between in sinister chief a decrescent Argent and in dexter chief a sun in splendour Or; in base seven castles Gules 3 and 4 each with two windows, with closed gates. 5. (enty in point) coat of arms of Dobrudja: on blue two golden dolphins facing downwards. Above all a shield of Hohenzollern. The collar of the Order of Carol I was also added (Înaltul Decret Regal 1921). This coat of arms continued to be used until 1947.

### 2.4. Bulgaria

Bulgaria was created in 1878 and the coat of arms of Bulgaria was taken from the Zhefarovic Stemmatographia. The Tarnovo Constitution of 1879, in article 21 defines: "The coat of arms of the Bulgarian state is a golden crowned lion on a crimson field. Above the field is a princely crown" (Конституцията на Българското княжество 1879). Two years later, two golden lions were added as supporters, including two national flags, a compartment, a motto, and mantling. Since the Constitution did not provide details and

there was no standardized source code, various variations were used over several decades (Figure 8a).

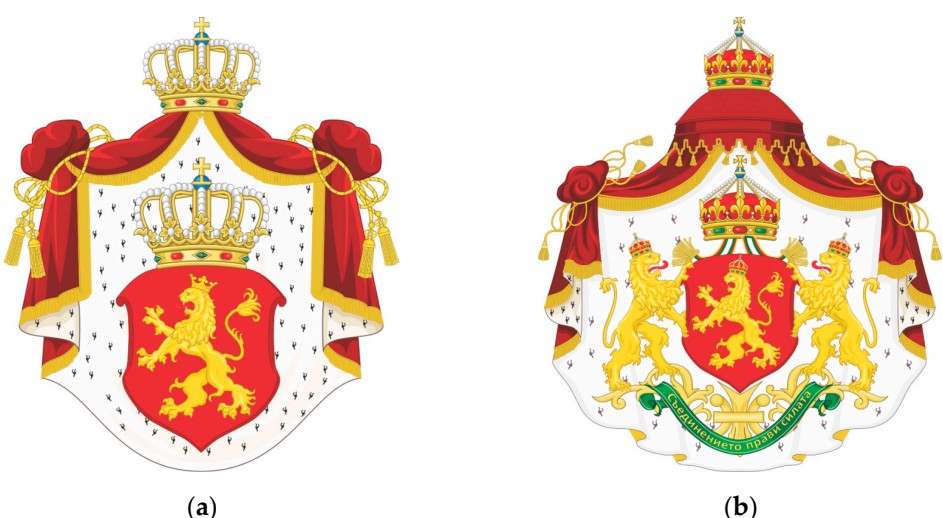

<table>
<tr><td>(<b>a</b>)</td><td>(<b>b</b>)</td></tr>
</table>

**Figure 8.** Coat of arms of Bulgaria: (**a**) 1878; (**b**) 1881–1927.

To solve this problem, a parliamentary commission was formed in 1923, which, only in 1927, made a decision, according to which the description says: the coat of arms of Bulgaria consists of a golden rampant crowned lion on a dark red shield; above the shield, a Bulgarian historical crown (Figure 8b). The shield is held by two crowned golden lions; below the shield, there is a compartment in the shape of oak branches and a white ribbon with the national motto "Единството прави силата"(Unity makes strength) (Войников 2017, p. 200) The coat of arms was designed by Stefan Badzov, born in 1881 in Krushevo, Macedonia (Јоновски 2019a, p. 109).

*2.5. Albania*

The modern Albanian state was created after the First Balkan War with the Declaration of Independence of Albania by the Great National Assembly in Valona on 28 November 1912. The red flag with a black double-headed eagle, proposed by Ismail Qemal, was adopted as the symbol of the new state. It was based on the coat of arms of Castrioti(c) found in the Illyrian Armorials, as Or a double-headed eagle Sable, each head crowned, in chief on a point Azure star Or. In the London armorial, the field is Azure and in the coat of arms, it is Gules, while the eagle is Azure. George Kastriot Skanderbeg used a double-headed eagle that can also be seen on the seal of a document sent by him to Pope Pius II in 1459 (Varfi 2000, p. 119).

The first attempt for a coat of arms was the coat of arms of Prince Wilhelm of Wied (7. 3–3.9 1914) who reigned with the title Widi I, but also as Skanderbeg II. It was actually Prince Wilhelm's coat of arms, Or, a peacock Sable with a bordure compony Gules and Sable and a double-headed eagle as a supporter. The arms has a large mantle Gules lined ermine, with a crown. The author of the project is Emil Doepler (Latifi 2022) (Figure 9a).

When Zogu was president of Albania in 1926, a banner was used instead of a coat of arms: Gules, double-headed eagle Sable, on the chest a helmet of Skanderbeg Or (Figure 9b). When Zogu proclaimed himself King, Albania got its first real coat of arms. The design of the arms is from February 1928 by Emil Doppler. The coat of arms was Gules, a double-headed eagle Sable, with a large mantle Gules doubled ermine, surmounted by Skanderbeg's helmet (Latifi 2022) (Figure 10a).

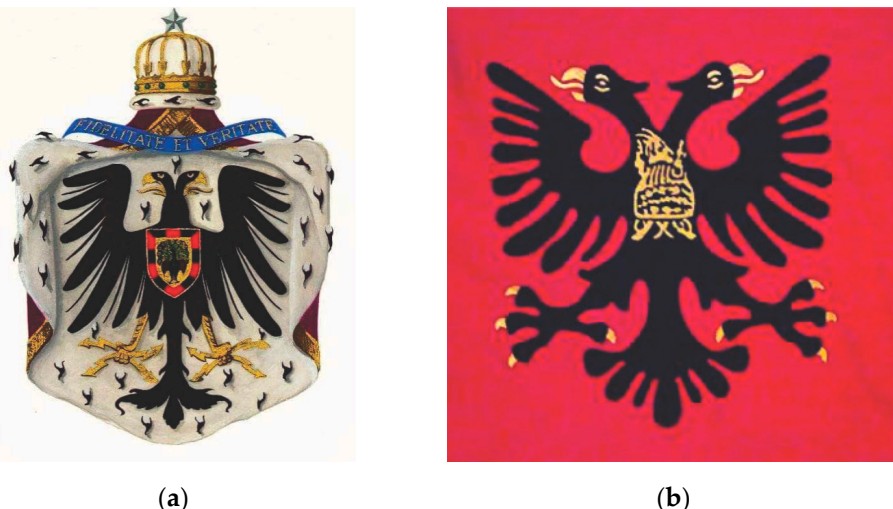

**Figure 9.** Coat of arms of Albania: (**a**) 1914; (**b**) 1926–1928.

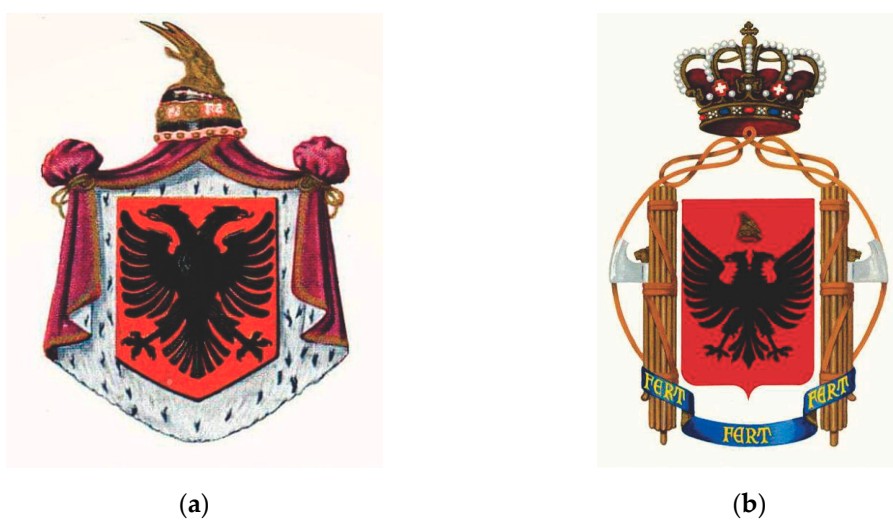

**Figure 10.** Coat of arms of Albania: (**a**) 1928–1939; (**b**) 1939–1944.

In 1939, when Italy ruled Albania, Skanderbeg's helmet from the great mantle gave way to the Italian crown, and it was placed above the shield, which in turn was supported by two fasces (Figure 10b).

### 2.6. Montenegro

The coat of arms of Montenegro originates from the middle of the 19th century, and slowly developed unofficially over a longer period (Маркуш 2007, pp. 21–29). It is based on the double-headed eagle from the coat of arms of Crnojevic from the Illyrian armorials, on which there is an escutcheon with a lion passant. Montenegro was officially recognised in 1878. The coat of arms was officially sanctioned for the first time by the first constitution of 1905, in article 39: "The coat of arms of the Principality of Montenegro is a white double-headed eagle with an imperial crown above the eagle's heads, with an imperial sceptre in the right and an orb in the left claw. On the breast is a lion on a red shield" (Устав за Књажевину Црну Гору 1905, p. 12). This design can be found on a medal from 1841. With the proclamation of the Kingdom of Montenegro, there is only one change to the coat of arms: the princely crown is replaced by a royal one (Figure 11).

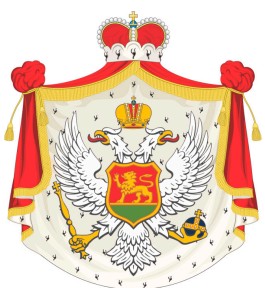

**Figure 11.** Coat of arms of Montenegro: 1878–1918.

The ruling Petrovic-Njegosh dynasty had its own coat of arms, which differed only in the field of the shield on the breast of the eagle, where it is blue with a green base on which stands a golden lion passant.

In 1918, Montenegro was abolished and incorporated into Serbia within the Kingdom of Serbs, Croats, and Slovenes, where Montenegro is no longer represented.

### 2.7. Slovenia, Croatia, and Bosnia and Herzegovina Immediately before and within the Kingdom of Serbs, Croats, and Slovenes

#### 2.7.1. Croatia

The coat of arms of Croatia, with the "chessboard"- Chequy Gules and Argent, has been used as an independent coat of arms of the pretentious coats of arms of the Habsburgs since 1495 (Božić and Ćosić 2021), and as a field from their arms of dominion since 1516, while on the territory of Croatia since the Charter of Cetingrad of 1527. It is also found in the Illyrian armorials and the Stemmatographia. In 1869 it was the coat of arms of Croatia, together with the coats of arms of Dalmatia and Slavonia on the coat of arms of the Triune Kingdom (Sbornik zakonah i naredabah valjanih za kraljevinu Hrvatsku i Slavoniju, kom. V. Zagreb 1868) (Figure 12a). In 1939, when Banovina Croatia was created, the coat of arms was the Yugoslav double-headed eagle Argent with the shield chequy on its breast (Okružnica Kabineta bana Banovine Hrvatske, br. 64178-1940. od 10. rujna 1940 1940) (Figure 12b).

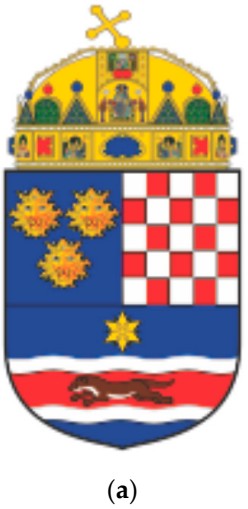

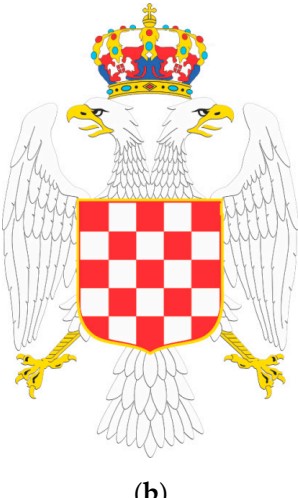

(**a**)                    (**b**)

**Figure 12.** Coat of arms of: (**a**) Triune Kingdom of Croatia, Dalmatia and Slavonia 1869; (**b**) Banovina Croatia 1939–1941.

#### 2.7.2. Bosnia and Herzegovina

After the annexation of Bosnia and Herzegovina by Austria-Hungary in 1878, a discussion developed about which was the "true coat of arms" of Bosnia and Herzegovina. In the next 10 years, studies of heraldic heritage followed. Historical coats of arms associ-

ated with Bosnia were two: Or, two staves in saltire embattled on the lower edge Gules each topped with a head Proper, crowned Or in Illyrian armorials and Gules, an arm in armour embowed fessways the hand flexed holding a scimitar Argent, which in the Illyrian Armorial is shown as the coat of arms of Rama; which, in turn was used as a coat of arms by Hungarian kings in the 16th century in Bosnia. The promotion of this coat of arms was led by the Hungarian historian, heraldist, and high-ranking civil servant Lajos Thallóczy, who wrote several publications on the matter. In them he defended the proposal for the coat of arms of Bosnia and Herzegovina to be exactly the coat of arms Or, an arm in armour embowed fessways the hand flexed holding a scimitar Argent. With that, he clearly defended the symbolic historical connection and Hungarian aspirations toward Bosnia and Herzegovina (Figure 13). It was his decision that the Land Government determined at the beginning of 1889 (Filipović 2020, pp. 173–75). Practically, it was used until 1918, when the territory of Bosnia and Herzegovina was included in the Kingdom of SCS, where it was not heraldically represented.

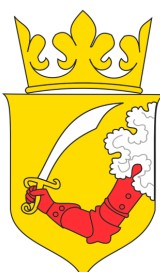

**Figure 13.** Coat of arms of Bosnia and Herzegovina 1978–1918.

### 2.7.3. Slovenia

Slovenia is a country with a rich heraldic heritage, with territorial coats of arms, including the provinces of Kranjska, Carinthia, Styria, but there was no coat of arms of Slovenia as a whole. During the First World War, the white-blue-red heraldic flag (first used in 1848) was used as the Slovenian flag, derived from the coat of arms of Kranjska, Argent, an eagle Azure armed Gules, with a crescent moon on the breast chequy Argent and Gules.

Slovenia, on the personal coat of arms of the King of the SCS, was represented on the third field with the coat of arms of Kranjska, but with inverted colours, Azure eagle Argent (Figure 14a), while on the state coat of arms, the coat of arms of Illyria (from the Illyrian coat of arms) was Azure, a star and in base a crescent Argent (Figure 14b) (Šišić 1920).

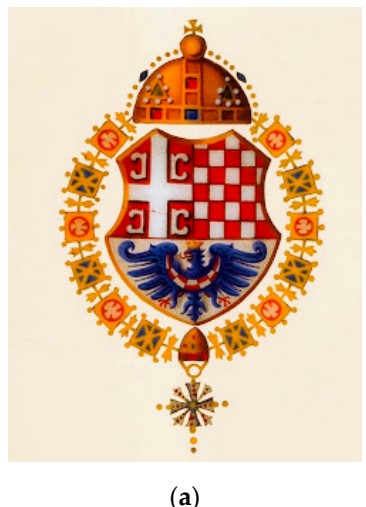    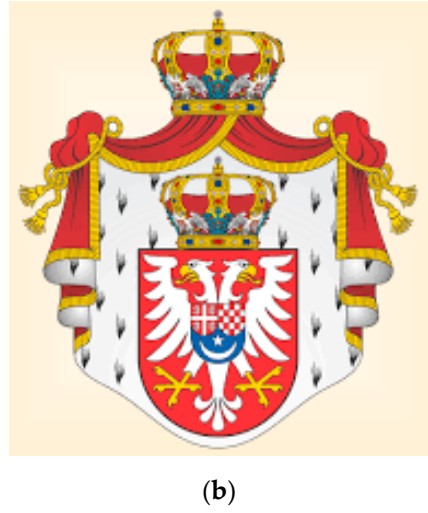

(**a**)                                                (**b**)

**Figure 14.** Coat of arms of Slovenia as third field in the arms of: (**a**) King of Serbians, Croatian and Slovenes (Yugoslavia); (**b**) Kingdom of Serbians, Croatian and Slovenes (Yugoslavia) 1918–1921.

In 1921, Slovenian MP Anton Sušnik asked to insert in the Slovenian field the three stars from the coat of arms of the Counts of Celje, as rulers of almost the entire territory that fell into Slovenia (Hribovšek 2018).

With the Vidovdan constitution of 28 June 1921 (Ustav and Slovenaca 1921), the Argent five-pointed star on the third field was replaced by three six-pointed stars Or. Instead of following the arrangement of the stars from the Celje coat of arms, where they are placed 1-2, in the Slovenian partition they are placed fesswise. On 14 October 1922, a decree was issued to change the colour of the crescent to chequy Argent and Gules, as in the coat of arms of Kranjska, but this was never implemented (Hribovšek 2018).

In the following years, the Slovenes did not accept the assigned coat of arms and did not identify with it. The architect Jože Plečnik, first in 1929 in Ljubljana, erected a monument in which the three stars were placed 2-1. In 1934, on Bled, he presented the three fields of the coat of arms of the Kingdom of Yugoslavia as separate coats of arms, on that of Slovenia, he used a coat of arms with three mountains (an allusion to Mount Triglav) and a six-pointed star. Triglav, with a five-pointed star, became the symbol of the Liberation Front in 1941. The coat of arms motif with Triglav with three six-pointed stars 2-1 appeared as a sign in the title of the newspaper Slovenska Zaveza from 1942 (Hribovšek 2018).

### 3. Second Period (1945–1989)

All of the countries subject to this study, except Greece, became People's/Socialist Republics after the Second World War and this was reflected on their coats of arms. When they adopted the Soviet understanding of state emblems, they incorporated some elements of their historical coats of arms or replaced them with new, socialist ones. Some of them adapted their arms to the new conditions[4]. Thus, the rising sun was found on the socialist coat of arms of several socialist countries. The Kingdom of Yugoslavia was replaced by the Federal People's Republic of Yugoslavia where the "coat of arms", the state emblem, "represents a field surrounded by ears of corn. The classes below are connected by a ribbon with the date 29-XI-1943 written on it. Between the tips of the ears, there is a (red) five-pointed star. In the middle of the field there are five torches placed diagonally, whose flames merge into one flame" (Figure 15a) (Устав на Федеративна Народна Република Југославија. 31.01.1946, Службен лист на ФНРЈ, 10/1946, 01.02.1946 1946a) With the constitution of 1963, the number of torches increased to six (Figure 15b) (Устав на Социјалистичка Федеративна Република Југославија, 07.04.1963, Службен лист на СФРЈ, 14/1963, 10.04.1963 1963).

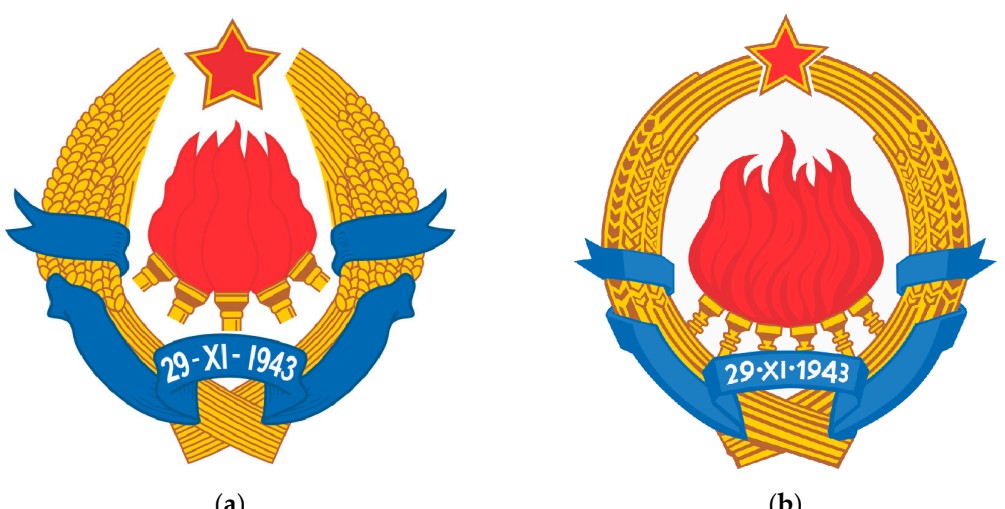

(a)          (b)

**Figure 15.** Socialist "coat of arms" of Yugoslavia (**a**) 1943–1963; (**b**) 1963–1992.

### 3.1. Greece

On 21 April 1967, a military junta overthrew King Constantine II, and abolished the monarchy. The coat of arms was, once again, a phoenix rising from the ashes (Figure 16a). In 1974, the military junta was abolished, as well as the monarchy, and the Third Greek Republic was proclaimed. On 7 June 1975, a new coat of arms was introduced, Azure a cross couped Argent, the shield surrounded by two laurel branches. This is a return to the traditional arms, but with a laurel wreath as the only external decoration. The government uses a stylised design by artist Kostas Grammatopoulos (Figure 16b) (Νόμος 851/21-12-1978 (ΦΕΚ 233 τ. Α) Περί εθνικής Σημαίας, των Πολεμικών Σημαιών καί του Διακριτικού Σήματος τού Προέδρου τής Δημοκρατίας 1978).

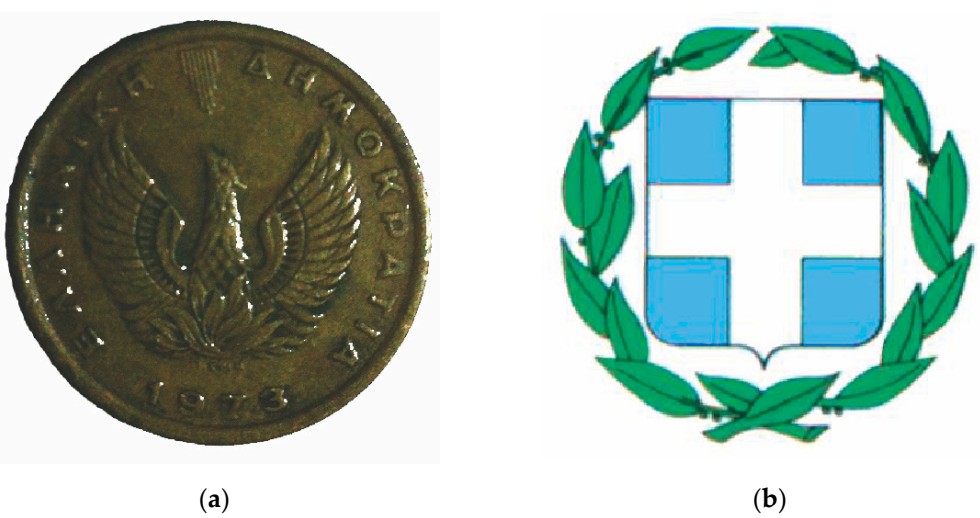

(**a**)          (**b**)

**Figure 16.** Coats of arms of Greece (**a**) 1967–1974 on a coin; (**b**) 1975-.

The constitution does not define the colour of the branches, but implies naturally coloured (green). The Greek government usually uses a monochromatic design in which the laurel branches are blue. A version with a gold laurel wreath is used by the military and on the presidential standard (The National Emblem n.d.).

### 3.2. Albania

After World War II, Albania's "coat of arms" had standard Soviet iconography—Gules a double-headed eagle Sable, the field surrounded by an ear of wheat, between the tips of which is a five-pointed star. Below the bundles are tied with red ribbon and dated 24 May 1944 (Figure 17) (Varfi 2000, p. 99).

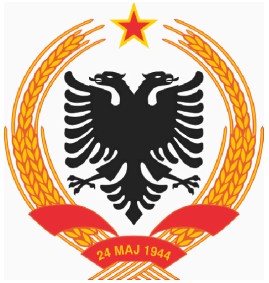

**Figure 17.** Socialist "coat of arms" of Albania 1946–1992.

### 3.3. Bosnia and Herzegovina

The first proposal for a "coat of arms" is found in the Proposal for the Constitution of the Republic of Bosnia and Herzegovina from 1946 and has a description: "The national coat of arms of the People's Republic of Bosnia and Herzegovina is a field surrounded by

an ear of corn. The ears below are connected by a ribbon with the date 1-VII-1944 written on it. There is a five-pointed star between the spikes. In the middle of the field is the outline of the Bosnian and Herzegovinian mountains, and in front of it is a torch held by three hands." (Nacrt Ustava NR BiH 1946, p. 2).

The adopted version differs in several details: the field is surrounded on the left by deciduous branches, and on the right by coniferous branches, which are connected by a ribbon at the bottom. In the field above the ribbon there are two factory chimneys and at the base lie two sheaves of grain. The silhouette of the city of Jajce is drawn in the background." (Figure 18). This coat of arms remained in force until 1992 (Filipović 2008, p. 110).

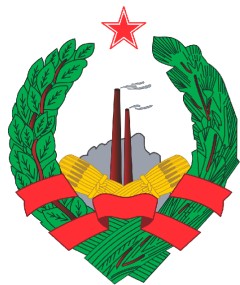

**Figure 18.** Socialist "coat of arms" of Bosnia and Herzegovina 1947–1992.

*3.4. Bulgaria*

After the abolition of the monarchy in 1946, the crown was removed from the coat of arms. Then, in January of 1948, the new socialist "coat of arms" of Bulgaria was set: Gules a lion Or, filed surrounded by ears of corn, and above his head there is a (red) five-pointed star (Figure 19a). Soon the field was changed to Azure (Войников 2017, p. 221). In 1971, the lion became silver. The state emblem had a red ribbon on which was written the date 9 September 1944, and later (1971), the year when the first Bulgarian state was created in the Middle Ages (681) was written on the ribbon (Figure 19b).

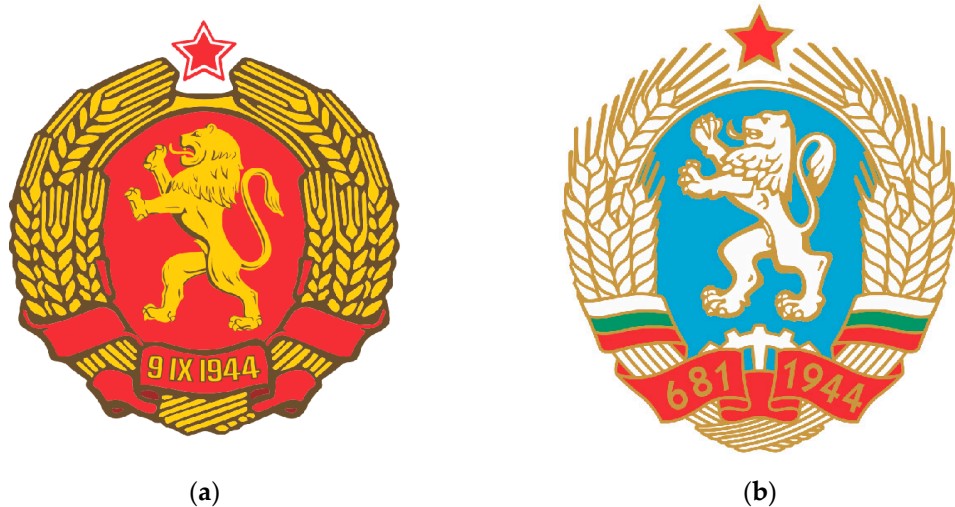

(**a**)　　　　　　　　　　　　　　　　(**b**)

**Figure 19.** Socialist "coat of arms" of Bulgaria (**a**) 1948; (**b**) 1971–1990.

*3.5. Macedonia*

The "coat of arms" of the Republic of Macedonia was a landscape composition, with a mountain as the central element, behind which the sun rises (Figure 20a). The authorship of the coat of arms is Vasilije Popovic—Cico. The drawing of the coat of arms of the law was used until the end of the same year (Jonovski 2009).

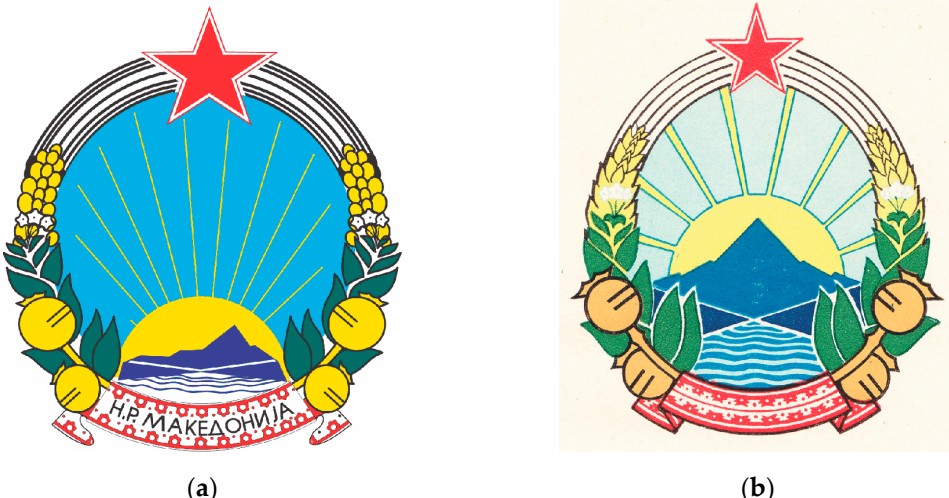

**Figure 20.** Socialist "coat of arms" of Macedonia (**a**) 1946; (**b**) 1947–2009.

The coat of arms of the Republic of Macedonia was adopted at the Second Extraordinary Session of the National Assembly, held in Skopje on 26 July 1946 (Народното собрание на Народна Република Македонија изгласа неколку закони важни за нашиот народ 1946).

"The Coat of Arms of the People's Republic of Macedonia is a field surrounded by stalks of wheat interwoven with fruits of opium poppy and tobacco leaves, which at the bottom are connected with a ribbon with folk design. The ribbon bears the text "N. R. Macedonia". Between the top of the stalks of wheat, there is a five pointed star. In the center of the field, a mountain is outlined, at the bottom of which a river is flowing. Behind the mountain, there is a sunrise" (Закон за грбот на Народна Република Македонија, Президиум на народното собрание на Народна Република Македонија 1946).

The coat of arms is a symbol of the freedom and brotherhood of the Macedonian people and the wealth of the Macedonian land. The red five-pointed star is a symbol of the national liberation war. The mountain and the river represent the Pirin and Vardar parts of Macedonia, which together represent "the unity of all parts of Macedonia and the ideal of our people for national unification" (Народното собрание на Народна Република Македонија изгласа неколку закони важни за нашиот народ 1946).

The "coat of arms" was redesigned by the end of the year and the text on the ribbon was omitted (Figure 20b) (Устав на Народна Република Македонија 1946c).

*3.6. Romania*

Romania completely abandoned the old coat of arms and adopted a short-lived socialist emblem with a tractor (Figure 21a). Within Article 99 of the 1948 Constitution, "the Coat of Arms of the Romanian People's Republic represents forests and mountains, over which the sun rises. In the middle is an oil well, and around it is a wreath of wheat ears" (Constituția Republicii Populare Române 1948). The 1952 constitution adds a red five-pointed star and a tricolour ribbon with the colours of the national flag and the letters R.P.R. (People's Republic of Romania) and it also defines the position of the oil well on the right (Constituția Republicii Populare Române 1952). With the constitution of 1965, the text of the ribbon was changed to "REPUBLICA SOCIALISTĂ ROMÂNIA" (Socialist Republic of Romania) (Figure 21b) (Constituția Republicii Socialiste România 1965).

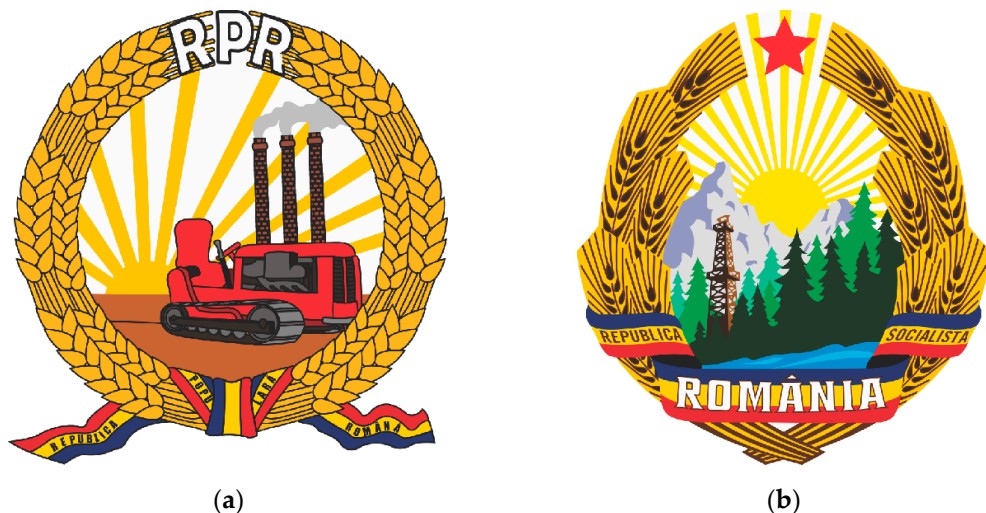

(**a**)                    (**b**)

**Figure 21.** Socialist "coat of arms" of Romania (**a**) 1948 (**b**) 1948 (1965)–1992.

### 3.7. Slovenia

After liberation, the coat of arms of the PR/SR of Slovenia was based on the symbol of the Liberation Front of the Slovenian People, Mount Triglav. It is surrounded by a wheat wreath interwoven with linden branches and a red ribbon. At the top is a red five-pointed star. The author of the design is Branko Simcic. The coat of arms is defined by Article 3 of the PR Slovenia Constitution of 17 January 1947 (Figure 22) (Ustavo Ljudske republike Slovenije 1947).

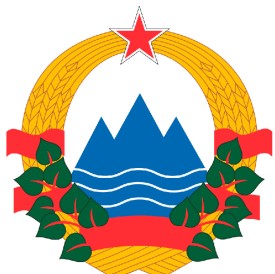

**Figure 22.** Socialist "coat of arms" of Slovenia 1947–1990.

### 3.8. Serbia

The People's Republic of Serbia acquired a new socialist coat of arms, which incorporated the historical coat of arms but without the cross. The 1947 Constitution defines it: "a field surrounded on one side by an ear of wheat, and on the other side by leafy oak branches. The wheat ear and oak branches are connected below by a ribbon on which are written the years of the national uprisings in Serbia, 1804 and 1941. Between the tops of the wheat ears and oak branches is a five-pointed star. In the field above the band is a sun, and in the sun is part of a gear. Above the sunbeams is a shield on which four fire steels are properly arranged'(Figure 23) (Устав Народне Републике Србије 1947).

Two years, 1804 and 1941, were written on the ribbon as years of uprisings against the occupiers. The grain represents the peasants, the cog the workers, and the rising sun, the new, brighter future. The design was by Gjorge Andrejević Kun (Круна води републику 2006).

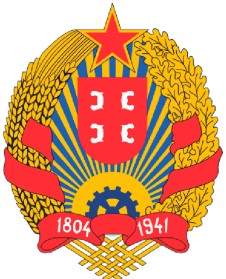

**Figure 23.** Socialist "coat of arms" of Serbia 1946–2004.

*3.9. Croatia*

In 1947, the traditional chequy coat of arms was incorporated into the socialist coat of arms of the People's Republic of Croatia "a field surrounded by two ears of grain in gold color. At the bottom of the field, there is an anvil over which the sea gently waves. From the sea rises the historic Croatian coat of arms, over which the sun rises. . .. Between the tops of the ears is a red five-pointed star trimmed with gold." (Figure 24) (Ustav Narodne Republike Hrvatske 1947) The chessboard started with Gules, as a sign of a return to tradition but also distancing itself from the coat of arms of the Independent State of Croatia (Zakonska odredba o državnom grbu, državnoj zastavi, Poglavnikovoj zastavi, državnom pečatu, pečatima državnih i samoupravnih ureda, 28. travnja 1941 1941), which, in turn, started with Argent (Čaldarović and Stančić 2011, p. 199).

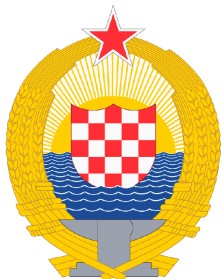

**Figure 24.** Socialist "coat of arms" of Croatia 1946–1990.

*3.10. Montenegro*

The first official symbol of federal Montenegro was adopted in 1945. The emblem shows Mount Lovchen with the church of Saint Peter Cetinje (chapel) and the tomb of Njegos atop, but without a cross on top, surrounded by a golden laurel wreath, while at the base there is a tricolour Montenegrin flag with the Cyrillic inscription Federal Montenegro, and a five-pointed star on top. The emblem was designed by Milan Bozović. With the Constitution of 31 December 1946, the inscription is omitted (Устав Народне Републике Црне Горе, 31.12.1946. Службени лист НР Црне Горе, 2/1947, 15.01.1947 1946b). The artistic representation of Lovchen with the church of Saint Peter Cetinje is also changed, as well as the red five-pointed star and the laurel wreath. The emblem was redesigned by Milo Milunović (Figure 25) (Маркуш 2007, p. 48).

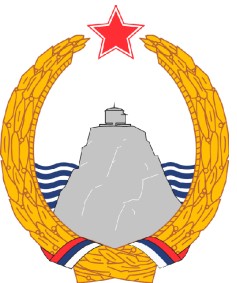

**Figure 25.** Socialist "coat of arms" of Montenegro 1947–1992.

## 4. Third Period (after 1990)

In the process of democratisation and the creation of new states, the question of new symbols that will express the new reality arose. Albania and Serbia-Montenegro have regained their old pre-war coats of arms. Bulgaria and Romania are the same, but with small changes to the symbols representing the monarchy. The newly created states of Bosnia and Herzegovina, Macedonia, Kosovo, Slovenia, and Croatia held a public competition.

The modern, democratic understanding of the choice of symbols may lead to the selection of symbols through public competition. Public competitions for state symbols are often conducted when, for various reasons, one needs to break with the historicity of the previous symbols and choose something new (Јоновски 2019b, p. 28). Thus, the "coat of arms" of Italy after the Second World War, in order to move away from the old symbols announced a competition to get a new symbol of the state. The competition stipulated that the design of the new emblem should have the Italian star, as an old symbol of Italy, without political and "historical" connotations. The emblem, with a socialist connotation, was adopted in May 1948 (The Emblem n.d.).

### 4.1. Albania

The Republic of Albania, on 7 April 1992, returned the small coat of arms from 1930 (Figure 26a). In 1998, Skanderbeg's golden helmet was placed on the shield in chief. To accommodate the helmet, the shield of the arms was elongated with a new ratio of 1:1.5 (Figure 26b) (Neni 7, Ligj per formen dhe permasat e Flamurit Kombetar permbajtjen e Himnit Kombetar formen dhe permasat e Stemes se Republikes se Shqiperise dhe menyren e perdorimit te tyre 2002).

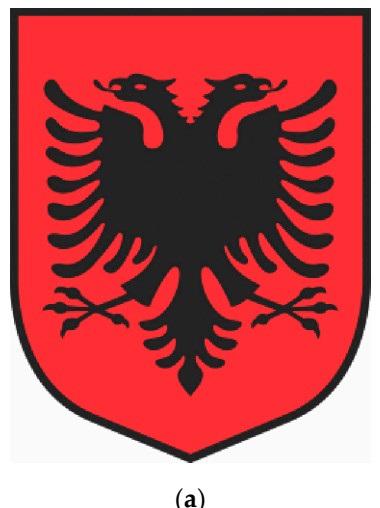

(**a**)

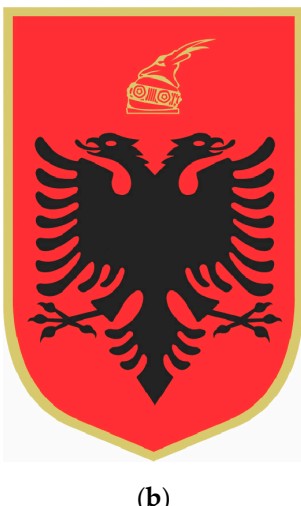

(**b**)

**Figure 26.** Coat of arms of Albania (**a**) 1992–1998; (**b**) 1998-.

### 4.2. Serbia

After the termination of the existence of the Federal Republic of Yugoslavia and the State Union of Serbia and Montenegro (Figure 27a), the working group for determining the symbols of Serbia, on 17 August 2004, recommended restoring the coat of arms of the Kingdom of Serbia from 1882. The coat of arms was only adopted on 19 May 2009 (Figure 27b) (Закон о изгледу и употреби грба, заставе и химне Републике Србије 2009). The original, with minor changes in the design of the coat of arms, was made by Ljubodrag Grujic and Dragomir Acovic, and was established on 11 November 2010 (Уредба о утврђивању оригинала великог и малог грба, оригинала заставе и нотног записа химне Републике Србије 2010).

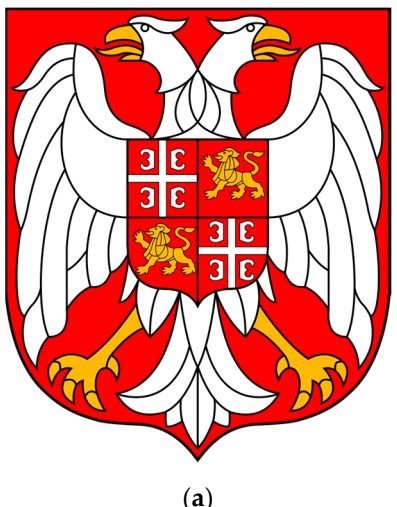 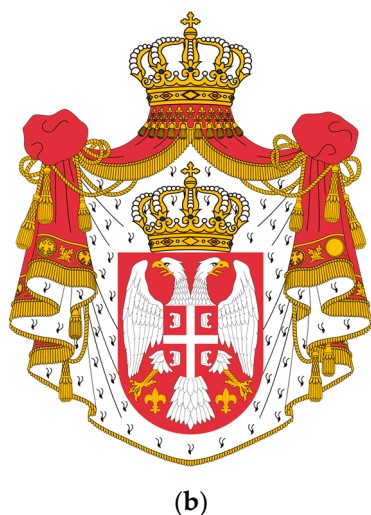

(**a**) (**b**)

**Figure 27.** Coat of arms of (**a**) State Union of Serbia and Montenegro 1993–2006; (**b**) Serbia 2004-.

### 4.3. Bulgaria

The Constitution of the Republic of Bulgaria from 1991, in article 164 stipulates "The coat of arms of the Republic of Bulgaria shows a golden rampant lion on a dark red shield" (Конституция на Република България 1991). For many years the design of the coat of arms was a source of great controversy, particularly the issue of the crown. Finally, on 4 August 1997 (Закон за Герб за Република България 1997), the coat of arms was adopted, which is actually the coat of arms of 1927, with minimal changes in the details of the crown, where the lilies have been replaced by crosses (Figure 28).

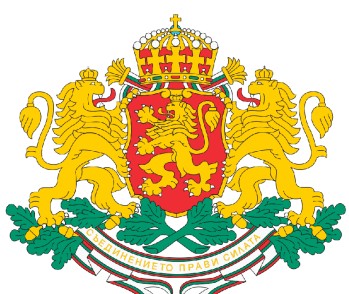

**Figure 28.** Coat of Arms of Bulgaria 1997-.

### 4.4. Romania

The revolution against Ceaușescu in 1989, was marked by the Romanian flag with a hole, from which the socialist coat of arms was cut. On 10 September 1992, the small coat of arms of 1922 was adopted for the coat of arms, but without the crown and the escutcheon of Hohenzollern (Figure 29a). The eagle's crown was reinstated in 2016 (Figure 29b) (Monitorul Oficial al României 2016).

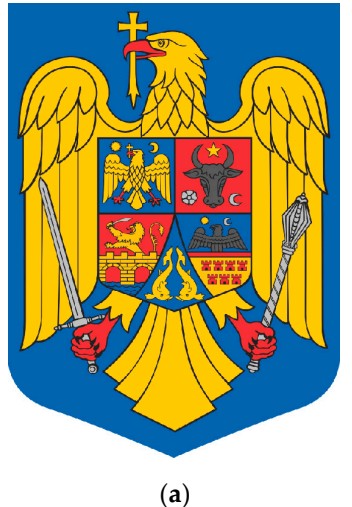
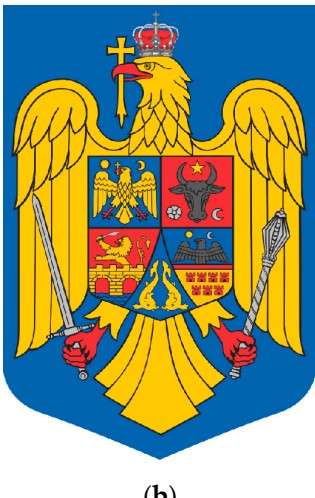

(**a**)　　　　　　　　　　　　　　　　　(**b**)

**Figure 29.** Coat of Arms of Romania (**a**) 1992–2016; (**b**) 2016-.

*4.5. Bosnia and Herzegovina*

On 27 February 1991, the Assembly passed a constitutional law regarding the state symbols of the Republic of Bosnia and Herzegovina, and a group for determining the symbols was formed. The multi-ethnic character dictated that the new state symbols be sought in the well-defined heraldic and vexillological symbols of Bosnia from the Middle Ages, with multi-ethnic unity, in which are the principles of peace and mutual tolerance of all the people of Bosnia and Herzegovina, regardless of nationality, religion or any other affiliation (Filipović 2008, p. 119).

Light blue was proposed for the flag, as a symbol of peace, with a ratio of 5:3, and the new coat of arms with proportions of 7:12 should be placed on it. The basis for the new coat of arms was the mantle of King Tvrtko I. Kotromanić, found during the excavation of the church where the Bosnian rulers were crowned and buried in Mili (today's Arnautovic) near Visoko. It was decided that the new state symbol should be the lily (heraldic lily) of the Bosnian subspecies (Lilium Bosniacum Beck).

The civil war interrupted the process, but Enver Imamović and Zvonimir Bebek continued the work on their own initiative. On 4 May 1992, it was established: "The coat of arms of the Republic of Bosnia and Herzegovina is in the form of a shield, blue, divided into two fields by a white crossbar with three golden-yellow lilies in each field." (Figure 30a) (Ustav Republike Bosne i Hercegovine (Prečišćeni tekst) 1993, p. 85).

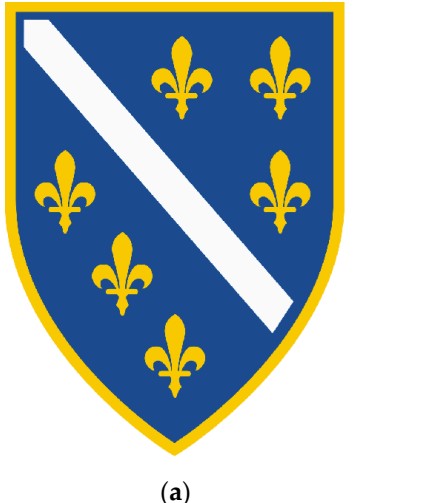
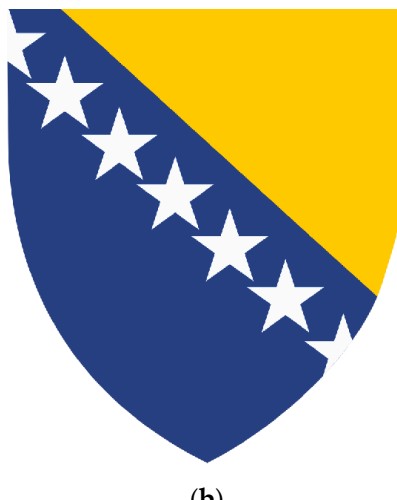

(**a**)　　　　　　　　　　　　　　　　　(**b**)

**Figure 30.** Coat of arms of Bosnia (**a**) 1991–1998; (**b**) 1998-.

The coat of arms on the flag was displayed on 21 May 1992, in front of the United Nations building in New York. But over time, the flag began to be associated only with Bosniaks. The flag, which was deliberately designed as a symbol of unity, has become a sign of division and discord (Filipović 2008, p. 119).

With the Dayton Peace Accords, new state symbols were needed. As the three parties failed to find a mutually acceptable solution, High Representative Carlos Westendorp appointed a seven-member commission "made up of people of good reputation who will guarantee a fair representation of the peoples of Bosnia and Herzegovina" in order to find a design for a new flag by the start of the Winter Olympic games in Nagano on 7 February 1998 (Filipović 2008, p. 119).

On 26 January, the Commission presented three designs with neutral geometric figures, acceptable to every citizen and every group in the country. All had a light blue background, like the flag of the United Nations. The yellow colour symbolized the sun as the source of light and life. The triangle can be associated with the geographical appearance of Bosnia and Herzegovina, and with the number of constituent nations that make it up. The large number of white stars symbolizes the European Union.

When MPs could not agree on a design, the High Representative imposed the flag with the most votes. The coat of arms is derived from the flag. The Law on the Coat of Arms of Bosnia and Herzegovina, in Article 4 states:

> "The coat of arms of Bosnia and Herzegovina is blue and in the form of a shield with a pointed end. There is a yellow triangle in the upper right corner of the shield. A row of white five-pointed stars runs parallel to the left side of this triangle." (Figure 30b) (Zakon o grbu Bosne i Hercegovine 1998, p. 351).

Two decades after their introduction, the state symbols are not accepted as true symbols of Bosnia and Herzegovina. Although proposed by a multi-ethnic commission, they were basically imposed by the High Representative. Despite the great propaganda of the international community, they failed to impose themselves on the entire territory of the country, where the old national symbols have the highest priority.

The Bosnian flag follows the concept of the Cypriot flag—the use of the contours of the country (albeit schematically), as a neutral national symbol (Filipović 2008, p. 126).

*4.6. Slovenia*

The Republic of Slovenia announced a public competition on 8 April 1991 for a coat of arms and flag. Out of 87 proposals, 4 finalists were selected that satisfied most of competition criteria, but the prizes were not awarded. But, instead of choosing from those who took part in the competition, it was decided on modified proposals of one of the finalists, Marko Pogažnik. So, the proposal bypassed the commission because the chosen version did not participate in the competition (Hribovšek 2018).

According to the official description, "The coat of arms of Slovenia is in the form of a shield. In the centre of the shield is depicted Mount Triglav, as an emblem in white on a blue background, with two wavy lines below it symbolising the sea and rivers, and three golden six-pointed stars arranged above it in the form of a downward-facing triangle. The shield has a red border on two of its sides. The coat of arms is designed in accordance with the set standard of geometry and colour" (Figure 31) (Zakon o grbu, zastavi in himni Republike Slovenije ter o slovenski narodni zastavi 1994).

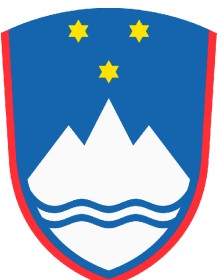

**Figure 31.** Coat of arms of Slovenia 1990-.

According to the Slovenian Heraldic Society, this "coat of arms" is actually an emblem (Hribovšek 2011).

*4.7. Macedonia*

The Republic of Macedonia also had a contest for a coat of arms and a flag. But the proposals that entered the further procedure had not even participated in the competition. Similarly, as in Slovenia, one of the three selected authors of the competition, Kostadin Tanchev Dinkata, was asked to propose a coat of arms and a flag with the same sun. It was with a sun with 32 rays. However, Todor Petrov's proposal for a Kutlesh-type (Vergina star) sun flag with 16 rays was adopted. The coat of arms of the same design was not adopted (Jonovski 2019). In addition to several attempts to establish a coat of arms with a lion, in 2009, the red five-pointed star was removed from the socialist coat of arms (Figure 32). In 2014, there was again a proposal with a lion, but it did not pass due to the greater political crisis that led to the Colourful Revolution.

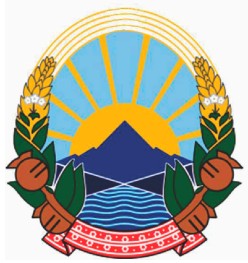

**Figure 32.** "Coat of Arms" of Macedonia 2009-.

*4.8. Kosovo*

The Constitutional Charter for Provisional Self-Government in Kosovo, promulgated by the United Nations Mission in Kosovo (UNMIK) in May 2001, stated that "The Provisional Institutions of Self-Government shall use only such symbols as are or as may be set forth in UNMIK legislation". It was established in 2003. The emblem of the Provisional Institutions for Self-Government depicted a map of Kosovo in gold on a blue background surrounded by two olive branches, in the style of those found used in the emblem of the United Nations, above which were three gold stars and three double-spirals ornamentation, which is a traditional symbol of ancient Dardnia and represents the rotating sun. (Figure 33a) (Implementing UNMIK Regulation No. 2001/9 on a Constitutional Framework for Provisional Self-Government in Kosovo 2003).

The Republic of Kosovo's choice of state symbols follows the pattern of symbols of a conflict. The Albanians of Kosovo used the flag, coat of arms, and anthem of the Republic of Albania, which they considered to be national rather than state symbols.

That is why new symbols had to be created, different from those perceived as national. The flag of the Republic of Kosovo was adopted immediately after the declaration of independence, on 17 February 2008. The flag is the result of an international public competition, organized by UNMIK, which attracted almost a thousand proposals. The competition conditions prohibited the use of the symbols previously used by the entities in

Kosovo, as well as the red–black and red–white combination. The winning design is by Muhamer Ibrahimi, and it is also used for the coat of arms of Kosovo. The coat of arms of Kosovo has a blue field, on which the map of Kosovo is represented, on which six stars are arranged in an arc, allsilver. The stars officially symbolize the six main ethnic groups in Kosovo (Albanians, Serbs, Turks, Gorani, Roma, and Bosniaks).

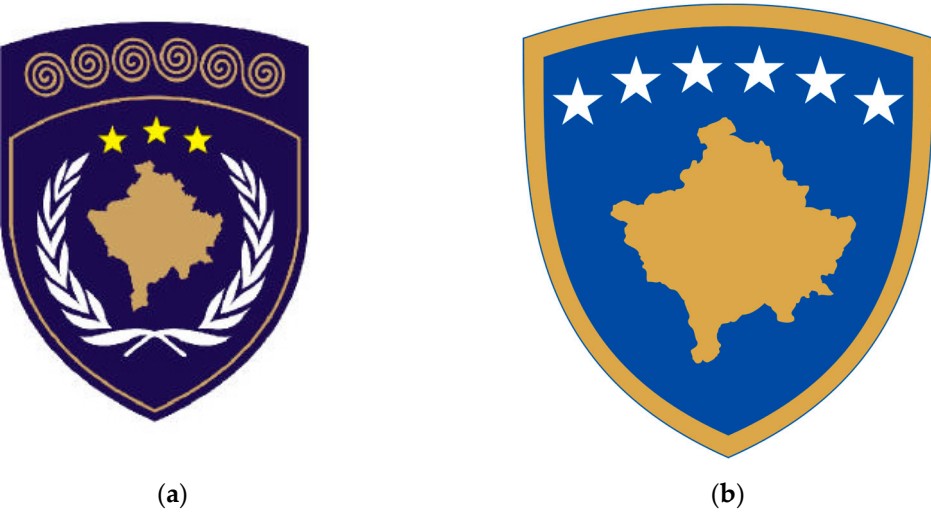

(**a**)  (**b**)

**Figure 33.** Coat of arms of Kosovo: (**a**) 2003–2008; (**b**) 2008-.

The Law on the Use of the State Symbols of Kosovo dated 20 February 2008, in Article 3, states that when adopted, the symbols will be an integral part of this law. No description is given of the coat of arms (Figure 33b) (Ljigji nr. 03/l-038 për Përdorimin e simboleve shtetërore të Kosovës, Gazeta zyrtare e Republikës së Kosovës/prishtinë: viti iii/nr 2008).

*4.9. Croatia*

On 29 June 1990, the constitution of the Socialist Republic of Croatia was changed, and the coat of arms is defined as the historical coat of arms of Croatia of 25 red and white fields (Figure 34a) (Odluka da se pristupi raspravi o promjeni Ustava Socijalističke Republike Hrvatske 1990). On 25 July, the amendments were adopted (Odluka da se pristupi raspravi o promjeni Ustava Socijalističke Republike Hrvatske 1990). Various versions of the checkboard arms were used in public spaces. Miroslav Šutej was appointed to redefine the coat of arms; he prepared designs for the coat of arms of Croatia, which, in addition to the checkerboard with the first red field, also includes regional coats of arms in the so-called republican tiara, according to the idea of Nikša Stančević, who, on the other hand, together with Zarko Domljan and the President of Croatia Franjo Tuđman, reviewed the proposals (Figure 34b). The final proposal was adopted by Parliament on December 21.

It was this unique design, later called the Croatian crown, that became a recognizable part that even inspired other coats of arms (Heimer 2008, p. 55). It is defined as "a crown with five spikes that joins the left and right upper parts of the shield in a slight arc. The crown holds five smaller shields with historical Croatian coats of arms from left to right in this order: the oldest known coat of arms of Croatia, the coats of arms of the Republic of Dubrovnik, Dalmatia, Istria, and Slavonia." (Stančić 2007, p. 5; Zakon o grbu, zastavi i himni Republike Hrvatske, te zastavi i lenti Predsjednika Republike Hrvatske 1990).

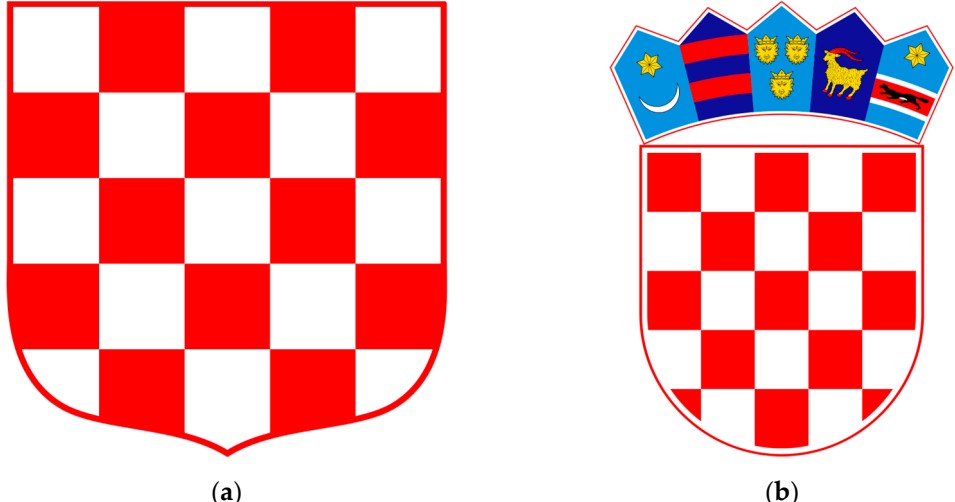

**Figure 34.** Coat of arms of Croatia (**a**) 1990; (**b**) 1991-.

*4.10. Montenegro*

The coat of arms of Montenegro was adopted in 1993 with the Law the Coat of Arms and Flag of Montenegro, which affirms the heraldic heritage of the Kingdom of Montenegro, using the coat of arms but without the shield (Figure 35a). Thus, the silver double-headed eagle becomes a supporter, which carries on its breast a red shield with a golden lion passant.

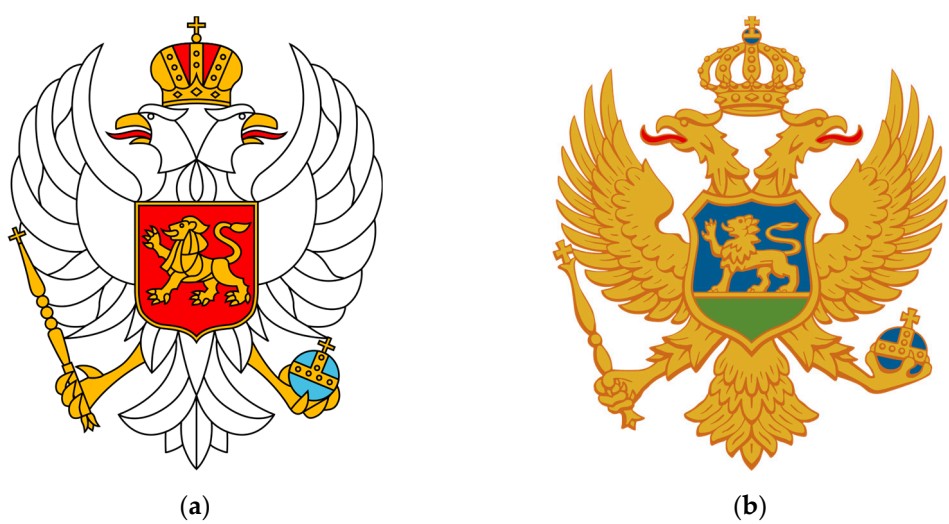

**Figure 35.** Coat of arms of Montenegro (**a**) 1992–2004; (**b**) 2004-.

The shield was also found in the second and third quarters of the coat of arms of the Federal Republic of Yugoslavia (Закон о употреби заставе, химне и грба Савезне Републике Југославије, Службени лист СРЈ 1993) designed by Bogdan Krsic (Figure 26a).

On 12 July 2004, a new law on the coat of arms of the Republic of Montenegro was adopted when the eagle became Or and the escutcheon became Azure, on a base Vert a lion passant Or, taken from the royal coat of arms of Montenegro (Figure 35b) (Маркуш 2007, p. 50).

**5. Analysis**

State heraldry in the Balkans began in the 19th century. Although a state emblem was first sanctioned by Greece in 1822, it is not compatible at all with heraldry and, thus, is not a coat of arms. The first true heraldic state coat of arms in the Balkans is that of Greece

under Prince Otto of Bavaria from 1831. It combined a coat of arms derived from the flag and the dynastic coat of arms of Wittelsbach (Bavaria). In the same period, Serbia also used a coat of arms that was officially sanctioned by the Constitution of 1835. Since Serbia and Montenegro had no foreign princes brought to the throne, the dynastic coats of arms are derived from the state coat of arms. The trend of including the coat of arms of the ruling dynasty is in all other kingdoms in the Balkans.

The coats of arms from the first period (1821–1944) can be divided according to the motifs that indicate heritage: 1. with the revolutionary past (Greece), 2. with a noble family from the past (Montenegro—Crnojevic; Albania—Kastrioti), and 3. with the territory (Bulgaria, Romania, Serbia, Croatia, Bosnia, and Herzegovina).

The coats of arms of Romania differ from the others, in that it is a coat of arms of dominion with fields of the coats of arms of its constituent territories, which is a classic Dominion coat of arms. So, it started with two fields with the coats of arms of Wallachia and Moldavia and ended up with five fields (additionally Banat, Transylvania, and Dobruja).

In the second period (1944–1990) all (except that of Greece) obtained socialist "coats of arms" with the same circular shield with a wreath of grain (or other floral symbol) with a ribbon at the base and a red five-pointed star at the top. According to the content of the field, they can be divided into 1. with the historical coat of arms (Albania, Bulgaria, Serbia, Croatia), and 2. with socialist motifs (Bosnia and Herzegovina, Macedonia, Slovenia, Romania, Montenegro). Of those with socialist motifs, all contain a mountain motif (on the coat of arms of Bosnia and Herzegovina it should be the outline of the city of Jajce, but visually it looks like a rock). On the other hand, four of them contain a rising sun, a concept found in the coat of arms of the USSR and all its republics.

The third period (1990 to the present) is characterised by the return of the coat of arms from the first period. They can be divided into coats of arms that 1. have been completely or mostly returned (Albania, Bulgaria, Romania, Serbia, Croatia, Montenegro), 2. Remained with socialist motifs (Macedonia and Slovenia), and 3. "Conflict" coats of arms (Bosnia and Herzegovina and Kosovo).

It is specific about the coats of arms of the 1st group that they did not completely return to their old coats of arms at once, but went through various modifications: Albania, six years after the return of the coat of arms, added the Skanderbeg's helmet. Bulgaria debated the use of the crown for six years before returning it in a modified form. Romania returned the crown to the eagle's head after 24 years. Serbia redesigned its coat of arms two years after its introduction, Croatia added the republican tiara to the original historical checky coat of arms a year later. After 11 years, Montenegro replaced the ruler's personal coat of arms with the coat of arms of the Kingdom.

A specific way to obtain a coat of arms in this period is a public contest, which is used for the coats of arms under the 2nd and 3rd of the above division. It is specific for Slovenia and Macedonia that the contests are only for a legal framework, because no proposal from the contest actually went to the next stage, but one of the finalists was chosen to implement the informal requests of the commission, which practically results in a custom-made coat of arms.

In the process of choosing the coat of arms of Macedonia, which began with the contest in 1992, there was a proposal for the sun first with 32 arms and then with 16 arms, Kutlesh type, designs the same as the flag. Both proposals were not adopted. If the last proposal, which was initially agreed upon, had been adopted, Macedonia would have received a coat of arms taken from the flag, just like Greece and Albania. With that, the flag would have become a heraldic standard (banner).

On the other hand, both Bosnia and Herzegovina and Kosovo have the same coat of arms concept that uses the design from the flag. Because they use the concept of neutral symbols with a geographical designation that originates from the flag of Cyprus, where the main emblem of the flag is the silhouette of the territory (map), while the olive branches are an additional element. In the coat of arms of Kosovo, the primary emblem is its map in white, while in Bosnia and Herzegovina, it is stylised in a yellow triangle. The field is

blue, derived from the flag of the United Nations, but also from the peace missions in those conflict countries, as well as from the flag of the European Union. The secondary elements for both coats of arms are the white five-pointed stars which, in addition to the official meaning, refer to the aspiration to the EU.

Today, from the coats of arms of 11 Balkan states, seven are historical (63%), two are coats of arms of "conflict", one is socialist, and one "heraldised socialist". In historical coats of arms, the most common coat of arms is the eagle. In addition to Albania, three countries have an eagle with a shield on its breast, Serbia, Montenegro, and Romania. Then follow a lion, a cross, and a checkerboard. The shield colours are three Gules and three Azure and one Gules and Argent. The other four "modern coats of arms" all have an Azure field. In total, the field is mostly Azure (63%). Of the charges, five are Or, four are Argent (white), one is Sable, and one is without a charge (the Croatian chessboard is considered a field).

**Funding:** This research received no external funding.

**Institutional Review Board Statement:** Not applicable.

**Informed Consent Statement:** Not applicable.

**Data Availability Statement:** Not applicable.

**Conflicts of Interest:** The author declares no conflict of interest.

## Notes

[1] Here we should point out the difference in the understanding of the term "coat of arms" in English heraldry and in its use in Balkan languages and understanding, especially used by the laws referring to their national Emblems as a "Coat of Arms" regardless of if they are good heraldry, Socialist heraldry or no heraldry at all. Due to the lack of real Western heraldic practice (apart from Romania, Slovenia and Croatia), and especially lacking theoretical heraldry to deal with the different aspects of the usage of symbol, emblem, and coat of arms, the national emblem is referred to as a "Coat of arms". Furthermore, there is a full concept of "Socialist heraldry" that existed in the entire region for a half century. In this article we will use terms "state emblem" for the emblems with no heraldry at all, "coat of arms" from perspective of Balkan heraldry which may differ from the English one. Also, we will use "socialist 'coat of arms'" for the socialist emblem so-called coat of arms, by primary sources, Laws, and books on East European heraldry.

[2] Coats of arms are viewed as symbols by symbolic interactionism. Emblem in Balkan context could be used as equivalent to English "charge" for the figure on the shield.

[3] Heraldically right-hand, but left-hand for the observer

[4] Poland deviated least from its historical coat of arms, a silver eagle on a red shield, making only a small intervention by removing the crown. Czechoslovakia retained its historic coat of arms, a red and silver two-tailed lion, but instead of the crown, a red five-pointed star with a gold border has been found. The small shield on the lion's chest, with a double bishop's cross, which historically represented Slovakia, was replaced by a flame symbol owing to the religious connotation.

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
