# Peer review of "The Development of the State Emblems and Coats of Arms in Southeast Europe"

_genealogy, doi:10.3390/genealogy7030054_

Round 1

Reviewer 1 Report

The article is relevant to modern heraldic science, though there are some points seem to be strange, such as avoidance of ancient and medieval periods in interpretation of coat-of-arms and evaluation of influence varieties of other European states.    

Author Response

Thank you for your revision,

The paper is already lengthy and including all the info about ancient symbols and medieval periods of all the modern states will make this paper into  a monograph, which is not the intended format for the journal

Reviewer 2 Report

From a methodological point of view, the article is well written and well structured. The results are relatively clearly presented and the references have been properly cited. I have, however, identified several typing errors in the bibliography and list of used works, so I suggest that the author has one more careful look at everything, especially the dates of publications.

Throughout the paper there are a number of typing and grammatical errors that could be eliminated by a native speaking proofreader.

Author Response

Thank you for your review, the article will be proofread by a native English speaker